# Chronic kidney disease and associated factors among adult population in Southwest Ethiopia

**Kindie Mitiku Kebede[1], Dejene Derseh Abateneh[1,2]\*, Melkamu Beyene Teferi[1], Abyot Asres[1]**

**1** College of Medicine and Health Sciences, Mizan Tepi University, Mizan Teferi, Ethiopia, **2** Menelik II College of Medicine and Health Sciences, Kotebe Metropolitan University, Addis Ababa, Ethiopia

\* dejenieh@gmail.com

**Data Availability Statement:** All relevant data are within the manuscript and its Supporting Information files.

**Funding:** Funding for this study was made possible through grants offered by Mizan Tepi University,

## Abstract

In Ethiopia, data on the burden and determinants of chronic kidney disease (CKD) in the general population is limited. This community-based study was conducted to assess the burden and associated factors of CKD among adults in Southwest Ethiopia. The study was conducted from August 23, 2018-October 16, 2018. Study participants were selected using a random sampling method. A structured questionnaire was used to collect relevant data. Blood pressure and anthropometric indices were measured following standard procedures. About 5 ml of urine sample was collected and the dipstick test was performed immediately. A blood sample of 3-5ml was collected for serum creatinine and blood glucose level determination. The three commonest estimators of glomerular filtration rate and the National Kidney Foundation Kidney Disease Outcomes Quality Initiative were used to define and stage CKD. Data were analyzed using SPSS 21. Multivariable logistic regression was employed and p-value <0.05 was used to indicate statistically significant results. A total of 326 participants with a mean age of 39.9(SD±11.2) years were enrolled in the study. The proportions of female participants (59.8%) were relatively higher than male participants (40.2%). The mean eGFR using CKD-EPI, CG and MDRD was 124.34 (SD±23.8) mL/min/1.73m$^2$, 110.67(SD±33.0) mL/min/1.73m$^2$ and 131.29 (SD±32.5) mL/min/1.73m$^2$ respectively. The prevalence of CKD was 7.4% using CKD-EPI & MDRD and 8% using CG. Similar finding using CKD-EPI & MDRD may indicate that either CKD-EPI or MDRD can be used to estimate GFR in this study area. In the age and sex-adjusted logistic regression model, hypertension was significantly associated with CKD using CKD-EPI & MDRD and age ≥40 years old was significantly associated with CKD using CG. Behavioral characteristics and other traditional risk factors were not significantly associated with CKD in the current study. The prevalence of CKD was high in the study area. Only hypertension and age ≥40 years old were significantly associated with CKD. More of the increased prevalence of CKD in the current study remained unexplained and deserves further study.

Ethiopia with grant number MTURD/3268/18). The award was received by the author Mr. Kindie Mitiku Kebede. The funder had no role in study design, data collection and analysis, decision to publish, or preparation of the manuscript.

**Competing interests:** The authors have declared that no competing interests exist.

# Introduction

Chronic kidney disease (CKD) is among Non-communicable diseases (NCD) which cause significant morbidity and mortality globally [1]. A recent systematic review and meta-analysis estimated that the global pooled prevalence of CKD ranges from 11–13% and 13.9% prevalence in sub-Saharan Africa [2]. The epidemiology of CKD in sub-Saharan Africa is different from other regions that it mainly affects the economically productive young age group. The risk of CKD like HIV infection, hypertension, diabetes, and other infectious diseases are increasing in sub-Saharan Africa [2–5].

Although Ethiopia is among the countries that strive to achieve sustainable development goals 2016–2030, the progress towards the reduction of NCD associated deaths is limited. More than one third of, (39%) annual deaths are associated with NCD [6]. The burden of CKD is increasing and mainly associated with poor community awareness, insufficient data, and poor healthcare infrastructure [7]. In Ethiopia, studies that assess the prevalence and associated factors of CKD are very limited. The available studies are not representative because they are taken from high-risk groups, like, diabetes and HIV patients [7, 8]. A high prevalence of CKD, 18.2% and 23.8% using the Modification of Diet in Renal Disease (MDRD) and Cockcroft-Gault (C-G) equations respectively was reported among diabetic patients [7]. Owing to limited community based published reports available so far on CKD in Ethiopia, this study was conducted to assess the prevalence and associated factors of CKD among adults in Southwest Ethiopia.

# Materials and methods

## Study area, study design, and participants

A community-based cross-sectional study was conducted in the Sheka zone, Southern Nation, Nationality and People Region (SNNPR), Southwest Ethiopia. It is bordered on the South by Bench Maji, on the West by the Gambella region, on the North by the Oromia region and on the East by the Kaffa zone. The administrative center of Sheka is Masha which is located approximately 610 Km Southwest of Addis Ababa. Based on the 2007 census conducted by the central statistical agency, the zone had a total population of 199,314 of whom 101,059 were men and 98,255 were women. About 34, 2227 (17.17%) were urban inhabitants.

The study was conducted from August 23, 2018 to October 16, 2018. The minimum sample size (N) was determined by using single population proportion formula [N = $(Z \alpha/2)^2 P (1-P) /d^2$], where $Z_{\alpha/2}$ = the value under standard normal table at 95% level of confidence which is 1.96, prevalence P, 15.2% prevalence of CKD among urban adults [9], margin of error d, which was set at 4%. Including a 10% non-response rate, the final sample size was 384 adults. Permanent residents aged ≥18 years in the households of Masha and Tepi town were included in the study. However, pregnant women and participants with acute illness with fever during data collection were excluded. Mentally disabled and those unable to give response were also excluded from the study.

## Sampling procedure

A total of two urban areas of Sheka zone was selected randomly i.e. Tepi and Masha from a total of three urban areas (Tepi, Mash and Andracha). Tepi town had three kebeles, while Masha town had two kebeles during the study period. A total of 3 kebeles were randomly included in the study from a total of 5 kebeles in the selected urban areas. Then, the lists of households were taken from the administration of each kebele. All households in the selected urban kebeles with adults ≥18 years old were enumerated with the help of health extension

workers to generate the sampling frame. After the generation of the sample framing, the sample size was allocated proportionally to the total household of each kebele. Finally, eligible participants were selected using a random sampling method. In the case of more than one adult population in a single household, one was selected using the lottery method. In case where eligible respondents were not available at the time of data collection, a revisit for three times was made. Respondents who were not available after three visits were considered as non-respondents.

## Data collection and laboratory methods

A structured questionnaire adapted from various literatures [9–11] was used to collect data on socio-demographic, economic, behavioral and co-morbidity related characteristics of participants.

Height, weight, blood pressure and waist circumference of all participants were measured by trained data collectors following standardized procedures. Blood pressure readings were performed using a digital blood pressure apparatus after the participant had remained in a sited position for three to five minutes on his or her right arm resting on the chair. Weight, height and waist circumference were measured in light clothing without shoe. Height was taken to the nearest 0.1 cm with a portable stadiometer (Seca 274, Germany) and weight was measured to the nearest 0.1k.g on mechanical Seca (Seca761, Germany). Waist circumference was measured using tape measures on the midway between the top of the hip bone and the bottom of the ribs. Participants were instructed to put off their heavy clothes and breathe out normally before waist circumference measurements. Waist circumference measurements were taken to the nearest 0.1cm. Blood pressure and all anthropometric measurements were made twice, and their averages were used in all analysis.

About 5 ml random urine sample was collected using a clean, leak-proof urine cup for urine chemical analysis. Chemical analysis of urine specimens was performed immediately after sample collection using urine dipsticks test (Multistix® Henry Schein, Inc. https://www.henryschein.com/medical-multistix.aspx). Semi-quantitative chemical analysis of CKD markers (protein) was performed.

Blood sample of 3–5 ml for serum creatinine and blood glucose level determination was collected using a syringe with needle and transferred to a gel containing serum separator test tube. Blood specimens were transported to the nearest general hospital (Tepi general hospital) for serum separation. A serum was separated after the sample clotted and centrifuged at 1000 – 2000g for 10 minutes by trained laboratory technologist. Serum sample was immediately separated from the whole blood and transferred to nunc tube. The serum was kept frozen at -20˚C until processed and then transported on ice-cooled containers to Jimma teaching and referral hospital, Ethiopia for analysis.

The creatinine and blood glucose levels were measured using the ABX pentra400 chemistry analyzer and reported in mg/dL or μmol/L (HORIBA, ABX, Japan). The ABX Pentra 400 is a multiparametric analyzer in which it has good reliability and practicability for routine and specialized clinical chemistry analyses, evaluated according to the National Committee for Clinical Laboratory Standards [12].

**Variables' definition.** The three common equations were used to estimate glomerular filtration rate (eGFR), and the National Kidney Foundation Kidney Disease Outcomes Quality Initiative (NKF/KDOQI) guidelines to define and stage CKD [13]. We defined a participant as having CKD if he or she had GFR below 60 ml/min/1.73 m$^2$ and/or had proteinuria (positive dipstick at least +1). Glomerular filtration rate classification was based on the following: stage 1: GFR > 90 with proteinuria; stage 2: GFR 60–89 with proteinuria; stage 3a: GFR 45–59; stage 3b: GFR 30–44; stage 4: GFR 15–29 and stage 5 end-stage renal disease (ESRD): GFR < 15 (Fig 1).

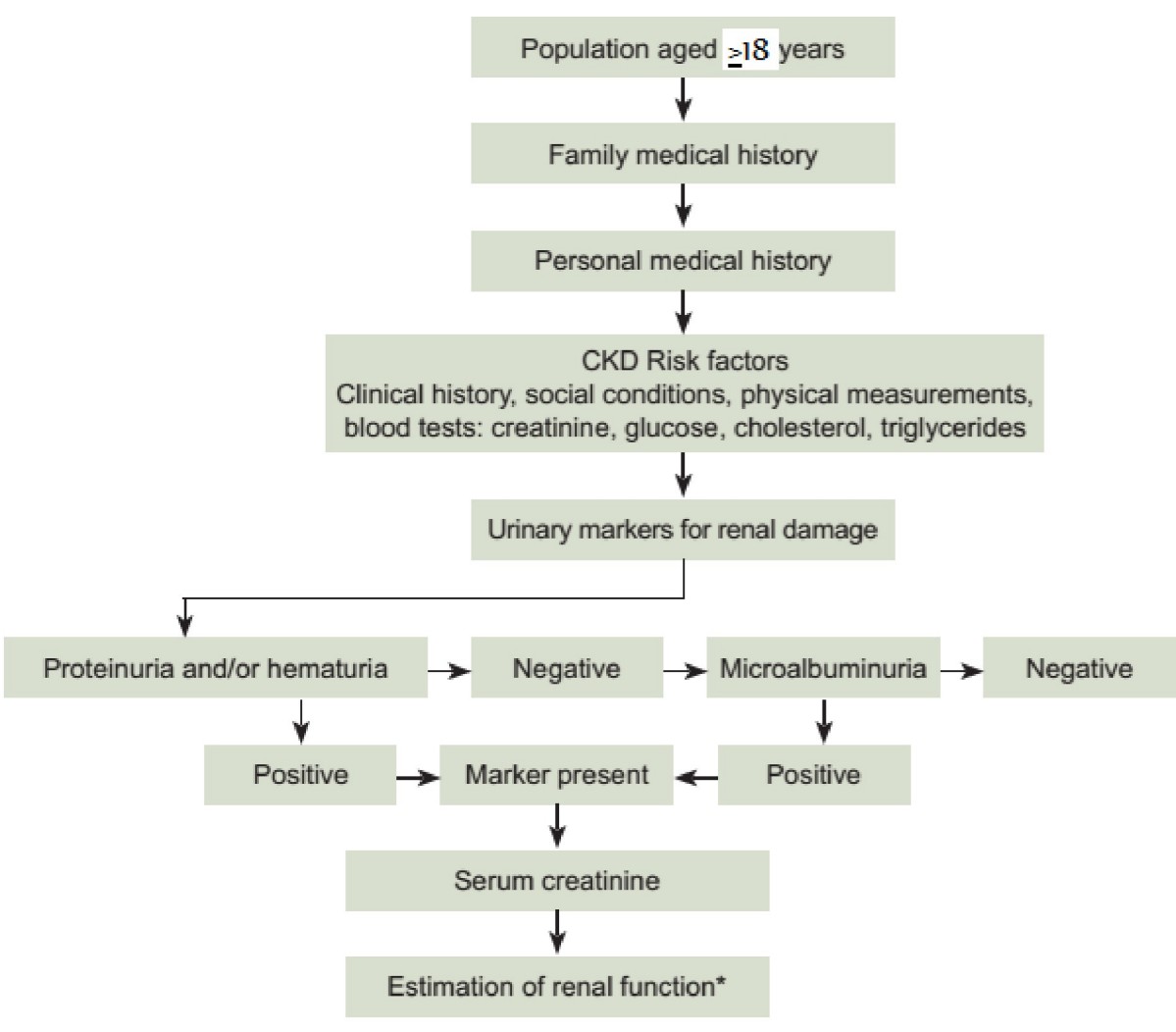

**Fig 1. Definitions and stages of chronic kidney disease [13].**

Hypertension was defined as a systolic (SBP) ≥140 mmHg and/or a diastolic blood pressure of (DBP) ≥90 mmHg or self-reported on anti-hypertension drugs. Diabetes was defined as random blood sager >200mg/dl and/or self-reported on anti-hyperglycemic medication. A separate cut of points was used to level central obesity for men and women: women were leveled as central obese if their waist circumference was ≥88 cm and men were leveled as central obese if their waist circumference ≥102 cm.

The nutritional status of participants was assessed using body mass index (BMI) and leveled as follows: underweight: <18.5; normal weight: 18.5–24.9; overweight: 25–29.9 and obese: >30.

**Traditional causes of CKD.**   Refer to the well-known causes of CKD such as hypertension, diabetes, obesity, and HIV.

**Heart diseases.**   In this study, it means a self-reported history of heart diseases such as coronary disease (vessel surgery), or heart failure, or heart attack or stroke.

**Moderate activities/exercise.**   Brisk walking or carrying light loads, gardening, household /domestic chores, roofing, painting for 30 minutes per day for five days a week.

**Vigorous exercise/activity.**   Carrying or lifting heavy loads, digging or construction work, running, fast cycling, aerobics, fast swimming, playing football & volleyball for 30 minutes per day for five days a week.

## Data analysis and interpretation

Data were entered into Epidata version 3.1 and exported to SPSS 21.0 for further analysis. Continuous data were presented as mean (± standard deviation) and categorical data as proportions. We used chi-square and fishers exact test for comparison of proportions and independent t-test for comparing mean differences. Logistic regression models were used to investigate the predictors of CKD. Variables with p-value of <0.2 in the bivariate analysis were included in the multivariable analysis. A p-values <0.05 were used to indicate statistically significant results.

## Data quality control

Data collectors were trained and supervised during the data collection process. The questionnaire was first prepared in the English language and translated to the local language (Amharic). About 10% of the questionnaire was pretested among urban Kebeles of Sheka zone which were not included in this study before the actual study period. Completeness and consistency of the collected data were checked during data collection by the supervisors. Anthropometric equipment's were calibrated frequently for every five measurements to control fallacy. Both intra-observer and interobserver variations were checked using SPSS version 21.0 during training of anthropometric measurement and the training was continued until an acceptable level of variation reached. Laboratory instruments were calibrated to guarantee reliability test result and laboratory tests were done according to manufacturer's instructions, with their respective controls.

## Ethics approval and consent to participate

Ethical approval was obtained from the ethics committee of Mizan Tepi University research directorate (Ref No: MTU/27/514/44/10). Permission of Sheka Zone, Southwest Ethiopia Health office and the head of the woreda were guaranteed. After explaining the objectives of the study, written consent was obtained from each study participant. For the participants who were not able to read and write, a fingerprint was used as a signature. The referral was considered for study participants with CKD and other medical risk factors.

## Results

### Socio-demographic characteristics

A total of 326 participants which gives a 95.9% response rate were included in this study. Most participants were within the age category of 18–39 (241; 73.9%) years. The proportions of female participants (59.8%) were relatively higher than male participants (40.2%). Significant

**Table 1. Socio-economic and demographic characteristics of study participants in Sheka zone, Southwest Ethiopia, 2018.**

| Characteristics | Overall | Men | Women | p-value |
|---|---|---|---|---|
| **Age** (Mean = 33.90 ± 11.2 years) | | | | 0.011 |
| 18–39 years old | 241(73.9) | 87(66.4) | 154(79.0) | |
| ≥ 40 years old | 85(26.1) | 44(33.6) | 41(21.0) | |
| **Education status** | | | | 0.163 |
| Not attended formal education | 53(16.3) | 17(13.0) | 36(18.5) | |
| Primary (1–8) | 116(35.6) | 42(32.1) | 74(37.9) | |
| Secondary (9–12) | 90(27.6) | 44(33.6) | 46(23.6) | |
| Above (12+) | 67(20.6) | 28(21.4) | 39(20.0) | |
| **Marital status** | | | | 0.032 |
| Married | 267(81.9) | 100(76.3) | 167(85.6) | |
| Others* | 59(18.1) | 31(23.7) | 28(14.4) | |
| **Religion** | | | | 0.477 |
| Christian | 242(74.2) | 100(41.3) | 31(36.9) | |
| Muslim | 84(25.8) | 142(58.7) | 53(63.1) | |
| **Occupation** | | | | <0.001 |
| Employed** | 53(16.3) | 22(16.8) | 31(15.9) | |
| Farming | 56(17.2) | 37(28.2) | 19(9.7) | |
| Merchant | 49(15.0) | 36(27.5) | 13(6.7) | |
| Housewife | 104(31.9) | 3(2.3) | 101(51.8) | |
| Students | 42(12.9) | 18(13.7) | 24(12.3) | |
| Others*** | 22(6.7) | 15(11.5) | 7(3.6) | |
| **Monthly income** ((Median 2000(IQR = 2200 EB)) | | | | 0 .110 |
| <2000EB | 113(34.7) | 38(36.9) | 75(46.9) | |
| ≥2000EB | 150(46.0) | 65(63.1) | 85(53.1) | |
| **Main source of water for drinking** | | | | 0.771 |
| Pip water | 171(52.5) | 70(53.4) | 101(51.8) | |
| Wall/spring | 155(47.5) | 61(46.6) | 94(48.2) | |

EB = Ethiopian birr*Single, divorced or widowed ** Private and government employed ***house maid, retired, unemployed

differences in age, marital status and occupation were observed between male and females (p<0.05) as indicated in Table 1.

## Behavioral and Lifestyle characteristics

Majority of participants didn't engage in vigorous activity/exercise (83.1%) or moderate activity/exercise (72.4%). A very small proportion (3.1%) of participants ever used herbal medicines. About 43 (13.2%) of the participants reported that they used nonsteroidal anti-inflammatory drugs. Nearly one-third of the participants (34.0%) reported that they drink <2 liters of water per day. A significant difference in alcohol and cigarette consumption, moderate exercise/activity, vigorous exercise/activity, and consumption of canned food between males and female were observed as described in Table 2.

## Clinical characteristics/Co-morbidities

One fifth (21.1%; n = 71) of the participants were hypertensive: defined as being on antihypertensive drug or having a systolic blood pressure of ≥140mmHg and /or diastolic blood pressure of ≥ 90mmHg. Similarly, one-fifth of the participants (20.2%; n = 66) reported history of

**Table 2. Behavioral related characteristics of study participants in Sheka zone, Southwest Ethiopia, 2018.**

| Characteristics | Overall | Men | Women | P-value |
|---|---|---|---|---|
| **Ever drink alcohol** | | | | <0.001 |
| Yes | 46(14.1) | 34(26.0) | 12(6.2) | |
| No | 280(85.9) | 97(74.0) | 183(93.8) | |
| **Ever smoke cigarette** | | | | |
| Yes | 17(5.2) | 17(13.0) | 0(0.0) | |
| No | 309(94.8) | 114(87.0) | 195(100.0) | |
| **Smoking year** | | | | 0.098 |
| <10 years | 13(76.5) | 49(37.4) | 91(46.7) | |
| ≥10 years | 4(23.5) | 82(62.6) | 104(53.3) | |
| **Herbal medicine intake** | | | | 0.326 # |
| Yes | 10(3.1) | 2(1.5%) | 8(4.1) | |
| No | 316(96.9) | 129(98.5) | 187(95.9) | |
| **Vigorous exercise/activity** | | | | 0.037 |
| Yes | 55(16.9) | 29(22.1) | 26(13.3) | |
| No | 271(83.1) | 102(77.9) | 169(86.7) | |
| **Moderate exercise/activity** | | | | 0.013 |
| Yes | 90(27.6) | 46(35.1) | 44(22.6) | |
| No | 236(72.4) | 85(64.9) | 151(77.4) | |
| **Added salt to food** | | | | 0.931 |
| Never | 18(5.5) | 8(6.1) | 10(5.1) | |
| Occasionally | 55(16.9) | 22(16.8) | 33(16.9) | |
| With each meal | 253(77.6) | 101(77.1) | 152(77.9) | |
| **Eat canned/processed food** | | | | <0.001 |
| Never | 193(59.2) | 63(48.1) | 130(66.7) | |
| Yes (occasionally and often) | 133(40.8) | 68(51.9) | 65(33.3) | |
| **Daily water intake** | | | | |
| <2 letters per day | 111(34.0) | 46(35.1) | 65(33.3) | 0.739 |
| ≥ 2 letters per day | 215(66.0) | 85(64.9) | 130(66.7) | |
| **Use of over NSAIDs drugs regularly** | | | | 0.002 |
| Yes | 43(13.2) | 8(6.1) | 35(17.9) | |
| No | 283(86.8) | 123(93.9) | 160(82.1) | |

# Fishers exact test was used

anemia. However, small proportion of the participants (3.4%; n = 11) were diabetic defined as being on drugs for diabetes and/or having blood glucose level >200mg/dl. Overall, there is a significant difference in the proportion of self-reported history of anemia and kidney stone between men and females (p<0.05) as described in Table 3.

## Anthropometric and laboratory measurements

Table 4 describes the mean differences of the anthropometric measurements and laboratory tests between men and women which were assessed using an independent t-test. A significant mean difference in height, BMI, serum creatinine, and eGFR using CKD-EPI and CG were observed between men and women (p<0.05). As expected, serum creatinine was higher in men than women (p < 0.001) but they had a significantly lower mean estimated creatinine clearance. We also found significantly high level of obesity among women as compared to men (p<0.001).

**Table 3. Clinical/co-morbidity related characteristics of study participants in Sheka zone, Southwest Ethiopia, 2018.**

| Characteristics | Overall | Men | Women | P-value |
|---|---|---|---|---|
| **Hypertension** | | | | 0.215 |
| Yes | 71(21.8) | 24(18.3) | 47(24.1) | |
| No | 255(78.2) | 107(81.7) | 148(75.9) | |
| **Diabetes** | | | | 0.761# |
| Yes | 11(3.4) | 5(3.8) | 6(3.1) | |
| No | 315(96.6) | 126(96.2) | 189(96.9) | |
| **Anemia** | | | | <0.001 |
| Yes | 66(20.2) | 6(4.6) | 60(30.8) | |
| No | 260(79.8) | 125(95.4) | 135(69.2) | |
| **Heart Disease**[*] | | | | 0.374 # |
| Yes | 12(3.7) | 3(2.3) | 9(4.6) | |
| No | 314(96.3) | 128(97.7) | 186(95.4) | |
| **Kidney stone** | | | | 0.049 |
| Yes | 34(10.4) | 19(14.5) | 15(7.7) | |
| No | 292(89.6) | 112(85.5) | 180(92.3) | |

[*] Heart Disease includes coronary disease (vessel surgery), heart failure, or heart attack or stork, # Fishers exact test was used

## Staging of kidney function and prevalence of chronic kidney disease

Using CKD-EPI estimator, about 23 (7.05%) of the participants had stage 1 and stage 2 CKD. Whereas, only 1 (0.3%) of them had chronic renal failure (stage 3a-5). The overall prevalence of CKD defined as proteinuria of $\geq$+1 and/or GFR<60 mL/min/1.73m$^2$ was 7.4% ((95% confidence interval (CI) [4.6–10.4]).

Using CG- estimator, about 21 (6.4%) of the participants had stage 1 and stage 2 CKD. Whereas, only 5 (1.5%) of them had chronic renal failure (stage 3a-5). The overall prevalence of CKD using CG-estimator and defined as proteinuria of $\geq$+1 and/or GFR <60 was 8.0% ((95% confidence interval (CI) [5.2–11.0]). whereas, using MDRD- estimator, about 23 (7.0%) of the participants had stage 1 and 2 CKD. Only 0.3% of them had chronic renal failure (stage 3a-5). The overall prevalence of CKD Using MDRD-estimator and defined as proteinuria of $\geq$+1and/or GFR<60 mL/min/ 1.73m$^2$ was 7.4% ((95% confidence interval (CI) [4.6–10.1]) as showed in Table 5.

**Table 4. Anthropometric and laboratory measurement results of study participants in Sheka zone, Southwest Ethiopia, 2018.**

| Characteristics | Overall | Men | Women | P-value |
|---|---|---|---|---|
| **Mean weight (Kg.)** | 60.96±9.63 | 61.49± 9.13 | 60.61±9.96 | 0.414 |
| **Mean height(meter)** | 1.60± 0.08 | 1.66±0.08 | 1.57±0.06 | <0.001 |
| **Mean SBP, mmHg (SD)** | 121.01±15.1 | 121.14±12.5 | 120.92±16.6 | 0.896 |
| **Mean DBP, mmHg (SD)** | 80.11± 10.7 | 79.30±9.2 | 80.66±11.6 | 0.262 |
| **Mean BMI, kg/m2 (SD)** | 23.84±4.10 | 22.52±3.7 | 24.73±4.1 | <0.001 |
| **Mean serum creatinine, mg/dl (SD)** | 0.74±0.22 | 0.99±.31 | 0.69±.14 | <0.001 |
| **Mean creatinine clearance, ml/min (SD)** | | | | |
| MDRD | 131.29±32.5 | 114.79±37.6 | 135.03±30.5 | 0.075 |
| CG | 110.67±33.0 | 90.20±32.6 | 115.32±32.7 | 0.033 |
| CKD-EPI | 124.34±23.8 | 107.10±33.3 | 128.26±19.5 | 0.010 |
| **Obesity** | | | | <0.001 |
| Yes | 25(7.7) | 2(1.5) | 23(11.8) | |
| No | 301(92.3) | 129(98.5) | 172(88.2) | |

**Table 5. Stages of kidney functions using the three eGFR estimators in Sheka zone, Southwest Ethiopia, 2018.**

| Stages | GFR estimation | CKD-Epi n (%) | CG n (%) | MDRD n (%) |
|---|---|---|---|---|
| 1 | $\geq$ 90, with proteinuria ($\geq$+1) | 19(5.8) | 17(5.2) | 20(6.1) |
| 2 | 60–89, with proteinuria ($\geq$+1) | 4(1.2) | 4(1.2) | 3(0.9) |
| 3a | 45–59.9, with or without proteinuria ($\geq$+1) | 1(0.3) | 4(1.2) | 1(0.3) |
| 3b | 30–44.9, with or without proteinuria ($\geq$+1) | 0(0) | 1(0.3) | 0(0) |
| 4 | 15–29.9, with or without proteinuria ($\geq$+1) | 0(0) | 0(0) | 0(0) |
| 5 | < 15, with or without proteinuria ($\geq$+1) | 0(0) | 0(0) | 0(0) |
|  | Overall CKD | 24(7.4) | 26(8.0) | 24(7.4) |

## Bivariate and multivariable analysis of associated factors

As the outcome variable is dichotomous, we applied logistic regression model. Before applying the model, the assumptions of logistic regression were checked. The linear relationship between continuous independent variables (age and income) and their logit transformation were checked. We found that there was no linear relationship between age and income and their log transformations. As a result, we included age and income as categorical variables in the model. We also investigated the multicollinearity between independent variables using the variance inflation factor. A significant multicollinearity was not identified. We used three separate logistic regression models for three common GFR estimators. However, the models explained only up 41% (Nagelkerke $R^2$) of the variance in the CKD. The Hosmer & Lemeshow test of the three models was not significant (p>0.05) which suggests good model fitness.

Using all the three GFR estimators, age, hypertension, heart disease, central obesity and use of over the counter NSAIDs drugs regularly were significantly associated with CKD in the Bivariate analysis (p<0.05). In the full adjusted model, only hypertension was significantly associated with CKD using CKD-EPI & MDRD- estimators described in Tables 6 and 7 respectively. On the other hand, age was significantly associated with CKD using CG- estimator in the full adjusted model as describe in Table 8. Using CKD-EPI & MDRD- estimators, participants who had hypertension were nearly 2.61 times more likely to have CKD than those who didn't have hypertension (p<0.001). Using CG- estimator, participants whose age ≥40 were 3.187 times more likely to exhibit CKD than whose age was 18–39 years (p = 0.005). However, central obesity and age showed borderline significant using CKD-EPI & MDRD estimators described in Tables 6 and 7 respectively.

## Discussion

In the current study, we assessed the prevalence and associated risk factors of chronic kidney diseases among the urban adult population in Sheka zone using the three commonest estimators of kidney functions. Over 7% of urban adults had CKD regardless of whether CKD-EPI, CG or MDRD GFR estimators were used. This finding is comparable with the prevalence of CKD reported in other sub-Saharan African counties; 10.4% in Nigeria [14], 12.4% in Kinshasa Congo [15], 10% in Cameroon [10], and 7% in Tanzania with 15.2% in the urban population [16]. The prevalence of CKD was slightly higher using the CG GFR estimator. It has been documented that CG overestimates the true prevalence of CKD [10, 16]. But, a similar prevalence of CKD was estimated using CKD-EPI and MDRD estimators. This may indicate that either CKD-EPI or MDRD can be used to estimate GFR in this study area. Other study findings, for instance, in Southern Africa and Ghana also showed that there is a high agreement between these two GFR estimators [17, 18].

**Table 6. Factors associated with CKD in age and sex adjusted logistic regressions using CKD-EPI estimator in Sheka Zone, Southwest Ethiopia, 2018.**

| Variables | Chronic Kidney Disease | | COR with 95% CI | P value | AOR with 95% CI |
|---|---|---|---|---|---|
| Socio-demographic characteristics | Yes (n, %) | No (n, %) | | | |
| **Sex** | | | | | |
| Male | 6(25.0) | 125(41.4) | 1 | | 1 |
| Female | 18(75.0) | 177(58.6) | 2.12(0.82–5.49) | 0.122 | 1.031(0.295–3.598) |
| **Age** | | | | | |
| 18–39 | 13(54.2) | 228(75.5) | 1 | | |
| ≥40 | 11(45.8) | 74(24.5) | 2.607(1.120–6.067) | 0.026 | 2.347(0.902–6.109) |
| **Education status** | | | | | |
| Not attended formal education | 1(4.2) | 52(17.2) | 0.642(0.277–1.492) | 0.303 | |
| Attended formal education | 23(95.8) | 250(82.8) | 1 | | |
| **Marital status** | | | | | |
| Married | 21(87.5) | 246(81.5) | 1 | | |
| Others | 3(12.5) | 56(18.5) | 0.628(0.181–2.177) | 0.463 | |
| **Religion** | | | | | |
| Christian | 17(70.8) | 225(74.5) | 1 | | |
| Muslim | 7(29.2) | 77(25.5) | 1.203(0.481–3.012) | 0.693 | |
| **Occupation** | | | | | |
| Employed* | 4(7.5) | 49(92.5) | | | |
| Farming | 2(3.6) | 54(96.4) | 0.454(0.080–2.587) | 0.374 | |
| Merchant | 4(8.2) | 45(91.8) | 1.089(0.257–4.613) | | |
| Housewife | 11(10.6) | 93(89.4) | 1.449(0.438–4.789) | | |
| Students | 2(4.8) | 40(95.2) | 0.613(0.107–3.518) | | |
| Others** | 1(4.5) | 21(95.5) | 0.583(0.061–5.535) | | |
| **Ethnicity** | | | | | |
| Sheka (natives) | 9(37.5) | 77(25.5) | 1.753(0.737–4.168) | 0.204 | |
| Others*** | 15(62.5) | 225(74.5) | 1 | | |
| **Monthly income** | | | | | |
| <2000EB | 9(45.0) | 104(42.8) | 1 | | |
| ≥2000EB | 11(55.0) | 139(57.2) | 0.914(0.366–2.287) | 0.848 | |
| **Main source of water for drinking** | | | | | |
| Wall/spring | 11(45.8) | 144(47.7) | 0.928(0.403–2.138) | 0.861 | |
| Pip water | 13(54.2) | 158(52.3) | 1 | | |
| Lifestyle/behavioral characteristics | | | | | |
| **Ever drink alcohol** | | | | | |
| Yes | 1(4.2) | 45(14.9) | 0.248(0.033–1.885) | 0.178 | 0.269(0.033–2.176) |
| No | 23(95.8) | 257(85.1) | 1 | | |
| **Ever smoke cigarette** | | | | | |
| Yes | 1(4.2) | 16 5.3% | 0.777(0.099–6.125) | 0.811 | |
| No | 23(95.8) | 286(94.7) | 1 | | |
| **Smoking year** | | | | | |
| <10 years | 0(0.0) | 13(81.3) | ≅0 | .999 | |
| ≥10 years | 1(100.0) | 3(18.8) | 1 | | |
| **Herbal supplement/ medicine** | | | | | |
| Yes | 2(8.3) | 8(2.6) | 3.341(0.669–16.696) | 0.142 | 0.997(0.150–6.621 |
| No | 22(91.7) | 294(97.4) | 1 | | |
| **Use of over the counter NSAIDs drugs regularly** | | | | | |
| Yes | 7(29.2) | 36(11.9) | 3.042(1.181–7.840) | 0.021 | 2.219(0.760–6.483 |

(*Continued*)

**Table 6.** (Continued)

| Variables | Chronic Kidney Disease | | COR with 95% CI | P value | AOR with 95% CI |
|---|---|---|---|---|---|
| **Socio-demographic characteristics** | **Yes (n, %)** | **No (n, %)** | | | |
| No | 17(70.8) | 266(88.1) | 1 | | |
| **Vigorous exercise /activity** | | | | | |
| Yes | 2(8.3) | 53(17.5) | 0.427(0.097–1.872 | 0.259 | |
| No | 22(91.7) | 249(82.5) | 1 | | |
| **Moderate exercise/activity** | | | | | |
| Yes | 9(37.5) | 81(26.8) | 1.637(0.689–3.887) | 0.264 | |
| No | 15(62.5) | 221(73.2) | 1 | | |
| **Eat canned/processed foods** | | | | | |
| Never | 13(54.2) | 180(59.6) | 1 | | |
| Yes (occasionally and often) | 11(45.8) | 122(40.4) | 1.248(0.542–2.878) | 0.603 | |
| **Added salt to food** | | | | | |
| Never | 1(4.2) | 17(5.6) | 0.871(0.109–6.970) | | 0.878(0.096–8.007) |
| Occasionally | 7(29.2) | 48(15.9) | 2.160(0.843–5.534) | 0.109 | 1.593(0.546–4.647 |
| With each meal | 16(66.7) | 237(78.5) | 1 | | |
| **Daily water intake** | | | | | |
| <2 letters per day | 11(45.8) | 100(33.1) | 1.709(0.739–3.951) | 0.210 | |
| ≥ 2 letters per day | 13(54.2) | 202(66.9) | 1 | | |
| **Co-morbidities/clinical characteristics** | | | | | |
| **Hypertension** | | | | | |
| Yes | 12(50.0) | 59(19.5) | 4.119(1.762–9.629) | 0.001 | 2.614(1.016–6.727 |
| No | 12(50.0) | 243(80.5) | 1 | | |
| **Diabetes mellitus** | | | | | |
| Yes | 1(4.2) | 10(3.3) | 1.270(0.156–10.357) | 0.824 | |
| No | 23(95.8) | 292(96.7) | 1 | | |
| **Anemia** | | | | | |
| Yes | 6(25.0) | 60(19.9) | 1.344(0.512–3.533 | 0.548 | |
| No | 18(75.0) | 242(80.1) | 1 | | |
| **Heart Disease***  | | | | | |
| Yes | 3(12.5) | 9(3.0) | 4.651(1.170–18.479) | 0.029 | 2.539(0.456–14.142 |
| No | 21(87.5) | 293(97.0) | 1 | | |
| **Kidney stone** | | | | | |
| Yes | 4(16.7) | 30(9.9) | 1.813(0.581–5.657) | 0.305 | |
| No | 20(83.3) | 272(90.1) | 1 | | |
| **Central obesity** | | | | | |
| Yes | 13(54.2) | 69(22.8) | 3.991(1.711–9.306) | 0.001 | 2.352(0.770–7.188 |
| No | 11(45.8) | 233(77.2) | 1 | | |
| **Nutritional status based on BMI** | | | | | |
| Under weight | 0(0.0) | 26(8.8) | ≅0 | | |
| Normal weight | 15(62.5) | 175(58.9) | 0.986(0.212–4.589) | | |
| Overweight | 7(29.2) | 73(24.6) | 1.103(0.214–5.684) | 0.907 | |
| Obese | 2(8.3) | 23(7.7) | 1 | | |

Using NKF/KDOQI staging, we found that most of the participants were in stage I & II. The finding is in line with several study findings in developing countries [9–11]. People within the early stage of CKD may not aware of their status. In fact, several studies showed that the majority of CKD patients don't aware their kidney problems [15, 19, 20]. Identification of

**Table 7. Factors associated with CKD in age and sex adjusted logistic regressions using MDRD estimator in Sheka Zone, Southwest Ethiopia, 2018.**

| Variables | Chronic Kidney Disease | | COR with 95% CI | P value | AOR with 95% CI |
|---|---|---|---|---|---|
| Socio-demographic characteristics | Yes (n, %) | No (n, %) | | | |
| **Sex** | | | | | |
| Male | 6(25.0) | 125(41.4) | 1 | | 1 |
| Female | 18(75.0) | 177(58.6) | 2.119(0.818–5.489) | 0.122 | 1.031(0.295–3.598) |
| **Age** | | | | | |
| 18–39 | 13(54.2) | 228(75.5) | 1 | | |
| ≥40 | 11(45.8) | 74(24.5) | 2.607(1.120–6.067) | 0.026 | 2.347(0.902–6.109) |
| **Education status** | | | | | |
| Not attended formal education | 1(4.2) | 52(17.2) | 0.209(0.028–1.582) | 0.303 | |
| Attended formal education | 23(95.8) | 250(82.8) | 1 | | |
| **Marital status** | | | | | |
| Married | 21(87.5) | 246(81.5) | 1 | | |
| Others | 3(12.5) | 56(18.5) | 0.628(0.181–2.177) | 0.463 | |
| **Religion** | | | | | |
| Christian | 17(70.8) | 225(74.5) | 1 | | |
| Muslim | 7(29.2) | 77(25.5) | 1.203(0.481–3.012) | 0.693 | |
| **Occupation** | | | | | |
| Employed* | 4(16.7) | 49(16.2) | | | |
| Farming | 2(8.3) | 54(17.9) | 0.454(0.080–2.587) | 0.374 | |
| Merchant | 4(16.7) | 45(14.9) | 1.089(0.257–4.613) | | |
| Housewife | 11(45.8) | 93(30.8) | 1.449(0.438–4.789) | | |
| Students | 2(8.3) | 40(13.2) | 0.613(0.107–3.518) | | |
| Others** | 1(4.2) | 21(7.0) | 0.583(0.061–5.535) | | |
| **Ethnicity** | | | | | |
| Sheka(natives) | 9(37.5) | 77(25.5) | 1.753(0.737–4.168) | 0.204 | |
| Others*** | 15(62.5) | 225(74.5) | 1 | | 1 |
| **Monthly income** | | | | | |
| <2000EB | 9(45.0) | 104(42.8) | 1 | | 1 |
| ≥2000EB | 11(55.0) | 139(57.2) | 0.914(0.366–2.287) | 0.848 | |
| **Main source of water for drinking** | | | | | |
| Wall/spring | 11(45.8) | 144(47.7) | 0.928(0.403–2.138) | 0.861 | |
| Pip water | 13(54.2) | 158(52.3) | 1 | | 1 |
| **Lifestyle/behavioral characteristics** | | | | | |
| **Ever drink alcohol** | | | | | |
| Yes | 1(4.2) | 45(14.9) | 0.248(0.033–1.885) | 0.178 | 0.269(0.033–2.176) |
| No | 23(95.8) | 257(85.1) | 1 | | 1 |
| **Ever smoke cigarette** | | | | | |
| Yes | 1(4.2) | 16(5.3) | 0.777(0.099–6.125) | 0.811 | |
| No | 23(95.8) | 286(94.7) | 1 | | |
| **Smoking year** | | | | | |
| <10 years | 0(0.0) | 13(81.3) | ≅0 | 0.999 | |
| ≥10 years | 1(100.0) | 3(18.8) | 1 | | |
| **Herbal supplements medicine** | | | | | |
| Yes | 2(8.3) | 8(2.6) | 3.341(0.669–16.696) | 0.142 | 0.997(0.150–6.621) |
| No | 22(91.7) | 294(97.4) | 1 | | 1 |
| **Use of over the counter NSAIDs drugs regularly** | | | | | |
| Yes | 7(29.2) | 36(11.9) | 3.042(1.181–7.840) | 0.021 | 2.219(0.760–6.483) |

(*Continued*)

**Table 7.** (Continued)

| Variables | Chronic Kidney Disease | | COR with 95% CI | P value | AOR with 95% CI |
|---|---|---|---|---|---|
| **Socio-demographic characteristics** | Yes (n, %) | No (n, %) | | | |
| No | 17(70.8) | 266(88.1) | 1 | | 1 |
| **Vigorous exercise /activity** | | | | | |
| Yes | 2(8.3) | 53(17.5) | 0.427(0.097–1.872) | 0.259 | |
| No | 22(91.7) | 249(82.5) | 1 | | |
| **Moderate exercise/activity** | | | | | |
| Yes | 9(37.5) | 81(26.8) | 1.637(0.689–3.887) | 0.264 | |
| No | 15(62.5) | 221(73.2) | 1 | | |
| **Eat canned/processed foods** | | | | | |
| Never | 13(54.2) | 180(59.6) | 1 | | |
| Yes (occasionally and often) | 11(45.8) | 122(40.4) | 1.248(0.542–2.878) | 0.603 | |
| **Added salt to food** | | | | | |
| Never | 1(4.2) | 17(5.6) | 0.871(0.109–6.970) | | 0.878(0.096–8.007) |
| Occasionally | 7(29.2) | 48(15.9) | 2.160(0.843–5.534) | 0.109 | 1.593(0.546–4.647) |
| With each meal | 16(66.7) | 237(78.5) | 1 | | 1 |
| **Daily water intake** | | | | | |
| <2 letters per day | 11(45.8) | 100(33.1) | 1.709(0.739–3.951) | 0.210 | |
| ≥ 2 letters per day | 13(54.2) | 202(66.9) | 1 | | |
| **Co-morbidities/clinical characteristics** | | | | | |
| **Hypertension** | | | | | |
| Yes | 12(50.0) | 59(19.5) | 4.119(1.762–9.629) | 0.001 | 2.614(1.016–6.727) |
| No | 12(50.0) | 243(80.5) | 1 | | 1 |
| **Diabetes** | | | | | |
| Yes | 1(4.2) | 10(3.3) | 1.270(0.156–10.357) | 0.824 | |
| No | 23(95.8) | 292(96.7) | 1 | | |
| **Anemia** | | | | | |
| Yes | 6(25.0) | 60(19.9) | 1.344(0.512–3.533) | 0.548 | |
| No | 18(75.0) | 242(80.1) | 1 | | |
| **Heart Disease*** | | | | | |
| Yes | 3(12.5) | 9(3.0) | 4.651(1.170–18.479) | 0.029 | 2.539(0.456–14.142) |
| No | 21(87.5) | 293(97.0) | 1 | | 1 |
| **Kidney stone** | | | | | |
| Yes | 4(16.7) | 30(9.9) | 1.813(0.581–5.657) | 0.305 | |
| No | 20(83.3) | 272(90.1) | 1 | | |
| **Central Obesity** | | | | | |
| Yes(obese) | 13(54.2) | 69(22.8) | 3.991(1.711–9.306) | 0.001 | 2.352(0.770–7.188) |
| No (not obese) | 11(45.8) | 233(77.2) | 1 | | 1 |
| **Nutritional status based on BMI** | | | | | |
| Under weight | 0(0.0) | 26(8.8) | ≅0 | | |
| Normal weight | 15(62.5) | 175(58.9) | 0.986(0.212–4.589) | | |
| Overweight | 7(29.2) | 73(24.6) | 1.103(0.214–5.684) | 0.907 | |
| Obese | 2(8.3) | 23(7.7) | 1 | | |

people in the early stage of CKD is important. This is because the cardiovascular risks associated with stage I & II is nearly equal to that of stage III [21].

After adjusting socio-demographic, lifestyle and co-morbidities related variables, only hypertension and advanced age were found to be significantly associated with CKD using

**Table 8. Factors associated with CKD in age and sex adjusted logistic regressions using CG estimator in Sheka Zone, Southwest Ethiopia, 2018.**

| Variables | Chronic Kidney Disease | | Unadjusted | P-value | Adjusted |
|---|---|---|---|---|---|
| **Socio-demographic characteristics** | Yes (n, %) | No n, %) | | | |
| **Sex** | | | | | |
| Male | 6(23.1) | 125(41.7) | 1 | | 1 |
| Female | 20(76.9) | 175(58.3) | 2.381(0.929–6.100) | 0.071 | 1.340(0.404–4.442) |
| **Age** | | | | | |
| 18–39 | 13(50.0) | 228(76.0) | 1 | | |
| ≥40 | 13(50.0) | 72(24.0) | 3.167(1.404–7.141) | 0.005 | 3.187(1.281–7.926) |
| **Education status** | | | | | |
| Not attended formal education | 2(7.7) | 51(17.0) | 0.407(0.093–1.776) | 0.232 | |
| Attended formal education | 24(92.3) | 249(83.0) | 1 | | 1 |
| **Marital status** | | | | | |
| Married | 23(88.5) | 244(81.3) | 1 | | 1 |
| Others | 3(11.5) | 56(18.7) | 0.568(0.165–1.959) | 0.371 | |
| **Religion** | | | | | |
| Christian | 18(69.2) | 224(74.7) | 1 | | 1 |
| Muslim | 8(30.8) | 76(25.3) | 1.310(0.547–3.135) | 0.544 | |
| **Occupation** | | | | | |
| Employed* | 4(15.4) | 49(16.3) | 1 | | |
| Farming | 3(11.5) | 53(17.7) | 0.693(0.148–3.256) | | |
| Merchant | 4(15.4) | 45(15.0) | 1.089(0.257–4.613) | 0.438 | |
| Housewife | 12(46.2) | 92(30.7) | 1.598(0.489–5.218) | | |
| Students | 2(7.7) | 40(13.3) | 0.613(0.107–3.518) | | |
| Others** | 1(3.8) | 21(7.0) | 0.583(0.061–5.535) | | |
| **Ethnicity** | | | | | |
| Sheka (natives) | 10(38.5) | 76(25.3) | 1.842(0.802–4.232) | 0.150 | |
| Others*** | 16(61.5) | 224(74.7) | 1 | | 1 |
| **Monthly income** | | | | | |
| <2000EB | 11(50.0) | 102(42.3) | 1 | | 1 |
| ≥2000EB | 11(50.0) | 139(57.7) | 0.734(0.306–1.758) | 0.488 | |
| **Main source of water for drinking** | | | | | |
| Wall/spring | 13(50.0) | 142(47.3) | 1.113(0.499–2.480) | 0.794 | |
| Pip water | 13(50.0) | 158(52.7) | 1 | | 1 |
| **Lifestyle/behavioral characteristics** | | | | | |
| **Ever drink alcohol** | | | | | |
| Yes | 1(3.8) | 45(15.0) | 0.227(0.030–1.715) | 0.151 | 0.253(0.032–2.025) |
| No | 25(96.2) | 255(85.0) | 1 | | |
| **Ever smoke cigarette** | | | | | |
| Yes | 1(3.8) | 16(5.3) | 0.710(0.090–5.578) | 0.745 | |
| No | 25(96.2) | 284(94.7) | 1 | | |
| **Smoking year** | | | | | |
| <10 years | 0(0.0) | 13(81.3) | ≅0 | 0.999 | |
| ≥10 years | 1(100.0) | 3(18.8) | 1 | | |
| **Herbal supplements /medicine** | | | | | |
| Yes | 2(7.7) | 8(2.7) | 3.042(0.611–15.132) | 0.174 | 1.106(0.167–7.342) |
| No | 24(92.3) | 292(97.3) | 1 | | |
| **Use of over the counter NSAIDs drugs regularly** | | | | | |
| Yes | 7(26.9) | 36(12.0) | 2.702(1.062–6.875) | 0.037 | 1.937(0.673–5.576) |

(*Continued*)

**Table 8.** (Continued)

| Variables | Chronic Kidney Disease | | Unadjusted | P-value | Adjusted |
|---|---|---|---|---|---|
| **Socio-demographic characteristics** | **Yes (n, %)** | **No n, %)** | | | |
| No | 19(73.1) | 264(88.0) | 1 | | |
| **Vigorous exercise /activity** | | | | | |
| Yes | 2(7.7) | 53(17.7) | 0.388(0.089–1.694) | 0.208 | |
| No | 24(92.3) | 247(82.3) | 1 | | |
| **Moderate exercise/activity** | | | | | |
| Yes | 10(38.5) | 80(26.7) | 1.719(0.749–3.944) | 0.201 | |
| No | 16(61.5) | 220(73.3) | 1 | | |
| **Eat canned/processed foods** | | | | | |
| Never | 15(57.7) | 178(59.3) | 1 | | |
| Yes (occasionally and often) | 11(42.3) | 122(40.7) | 1.070(0.475–2.409) | 0.870 | |
| **Added salt to food** | | | | | |
| Never | 1(3.8) | 17(5.7) | 0.768(0.097–6.104) | | 0.716(0.078–6.543) |
| Occasionally | 7(26.9) | 48(16.0) | 1.904(0.754–4.809) | 0.173 | 1.471(0.511–4.233) |
| With each meal | 18(69.2) | 235(78.3) | 1 | | |
| **Daily water intake** | | | | | |
| <2 letters per day | 11(42.3) | 100(33.3) | 1.467(0.650–3.311) | 0.357 | |
| ≥ 2 letters per day | 15(57.7) | 200(66.7) | 1 | | |
| **Co-morbidities/clinical** | | | | | |
| **Hypertension** | | | | | |
| Yes | 12(46.2) | 59(19.7) | 3.501(1.539–7.965) | 0.003 | 2.105(0.836–5.299) |
| No | 14(53.8) | 241(80.3) | 1 | | |
| **Diabetes** | | | | | |
| Yes | 1(3.8) | 10(3.3) | 1.160(0.143–9.433) | 0.890 | |
| No | 25(96.2) | 290(96.7) | 1 | | |
| **Anemia** | | | | | |
| Yes | 6(23.1) | 60(20.0) | 1.200(0.462–3.119) | 0.708 | |
| No | 20(76.9) | 240(80.0) | 1 | | |
| **Heart Disease*** | | | | | |
| Yes | 3(11.5) | 9(3.0) | 4.217(1.068–16.661) | 0.040 | 2.555(0.462–14.133) |
| No | 23(88.5) | 291(97.0) | 1 | | |
| **Kidney stone** | | | | | |
| Yes | 4(15.4) | 30(10.0) | 1.636(0.529–5.066) | 0.393 | |
| No | 22(84.6) | 270(90.0) | 1 | | |
| **Central Obesity** | | | | | |
| Yes | 14(53.8) | 68(22.7) | 3.980(1.758–9.011) | 0.001 | 2.147(0.748–6.162) |
| No | 12(46.2) | 232(77.3) | 1 | | |
| **Nutritional status based on BMI** | | | | | |
| Under weight | 1(3.8) | 25(8.5) | 0.460(0.039–5.418) | 0.537 | |
| Normal weight | 15(57.7) | 175(59.3) | 0.986(0.212–4.589) | | |
| Overweight | 8(30.8) | 72(24.4) | 1.278(0.253–6.451) | | |
| Obese | 2(7.7) | 23(7.8) | 1 | | |

CKD-EPI/MDRD & CG estimators respectively. However, more of the increased prevalence of CKD in this study area seems to be unexplained. In line with our study finding, a recent study finding in Tanzania and Uganda showed that the majority of the increase in the risk of CKD was not explained [9, 11]. Unexplained risk factors of CKD in this study area may suggest

that other non-traditional risk factors might contribute to the high burden of CKD. Some study findings in another part of the world showed that nontraditional risk factors such as heavy metals and agro-chemicals may contribute to CKD in tropical and semi-tropical areas [22, 23]. Some other studies suggested that infectious diseases such as HIV/AIDS, Schistosomiasis and Leishmaniasis are also associated with CKD [24–26]. However, as a limitation of this study, we didn't control the confounding effect of these risk factors of CKD. Large-scale epidemiological studies are needed to better understand the specific risk factors of CKD and, to examine the potential but unmeasured above risk factors.

As documented in the broad literature [7, 10, 11, 19], advanced age is a well-known risk factor of CKD. As age increases, there may be salt sensitization, hardening of arteries and scarring of the tiny blood vessels in the kidney. This may, in turn, enhance protein excretion and decrease GFR [27–29]. However, as many of the adult participants in the current study didn't have a birth certificate, they may underestimate their age. Underestimation of age could lead to overestimation of GFR using any of the methods used to calculate GFR. This may in turn underestimate the true prevalence of CKD in this study area since the level of creatinine is affected by socio-demographic factors such as age [30].

In the current study, among the traditional risk factors of CKD, hypertension was found to be significantly associated with CKD. This is in line with several community-based study findings [10, 11, 14, 20]. Hypertension may elevate intraglomerular pressure which results in glomerulosclerosis and eventual protein trafficking [31, 32]. However, as a limitation of this study (chicken egg dilemma), it is hard to distinguish whether hypertension causes CKD or CKD causes hypertension. An increased risk of CKD among hypertensive participants may warrant urban adults to adhere to treatment and control their blood pressures. Studies have shown that interventions that decrease blood pressure levels in patients with proteinuria delay progression to CKD [33, 34].

Co-morbidity related factors such as diabetes, anemia, kidney stone; heart disease, central obesity and nutritional status measured by BMI were not significantly associated with CKD. However, central obesity showed a borderline significant. It has been documented that waist circumference or waist to hip ratio predicts cardiovascular risks than BMI [35]. The low prevalence of the above-mentioned co-morbidities in our participants makes any comment of their association difficult to ascertain.

In the current study, behavioral factors such as alcohol intake and cigarette smoking were not significantly associated with CKD. This is in line with several study findings [14, 36, 37]. Alcohol intake and cigarette smoking are often considered anti-social habits in Ethiopia so people may give a negative response. This may, in turn, nullify the true association between alcohol intake, cigarette smoking and, CKD.

Overall, in this study area, CKD is significantly associated with age ≥40 years old and hypertension. Early screening of high-risk populations (elderly and hypertensive peoples in the case of this study area) may help to diagnose people at the early stage of CKD. This is very important as Ethiopian health care institutions may not be able to support the economic burden of providing dialysis and renal replacement therapy to all patients. Dialysis and renal replacement therapy are not free in Ethiopia. This may result catastrophic out of the pocket expenditure. Thus, early screening of high-risk populations to CKD is important to minimize complications and out of pocket expenditures.

This study was the first community-based assessment of CKD in Southwest Ethiopia using the three common GFR estimators. The finding of the study will provide information about the urban prevalence of CKD in Sheka zone. However, this study has several limitations. The cross-sectional nature of this study restricted us to infer causality. The common GFR measures used in this study were not validated in Ethiopia. The absence of reliable and validated

measures of kidney function tests has made difficult the estimation of CKD in Africa [16, 18]. This study exclusively focused on the urban population. Thus, the estimated CKD may not reflect the prevalence and risk factors of rural communities. Because of security and financial restrictions, we didn't make a three months control of positive findings as suggested by the KDIGO guideline. Some risk factors identified elsewhere such as heavy metals, agrochemicals, Schistosomiasis, Leishmaniasis and HIV [22–26] were not investigated in this study.

## Conclusions

The prevalence of CKD was high in the study area. Only hypertension and age ≥ years old were significantly associated with CKD. More of the increased prevalence of CKD in the current study remained unexplained and deserves further study. Policymakers and programmers need to design strategies and encourage high-risk populations to be screened as early as possible. It is better if clinicians should also consider kidney function marker tests for hypertensive and patients older than 40 years old.

## Supporting information

**S1 Questionnaire. Questionnaire in English and local language (Amharic).**
(PDF)

**S1 Data.**
(SAV)

## Acknowledgments

The authors would like to acknowledge all the data collectors, supervisors and study participants.

## Author Contributions

**Conceptualization:** Kindie Mitiku Kebede, Dejene Derseh Abateneh, Melkamu Beyene Teferi, Abyot Asres.

**Data curation:** Kindie Mitiku Kebede, Dejene Derseh Abateneh.

**Formal analysis:** Kindie Mitiku Kebede.

**Funding acquisition:** Kindie Mitiku Kebede.

**Investigation:** Kindie Mitiku Kebede, Dejene Derseh Abateneh, Abyot Asres.

**Methodology:** Kindie Mitiku Kebede, Dejene Derseh Abateneh, Melkamu Beyene Teferi, Abyot Asres.

**Project administration:** Kindie Mitiku Kebede, Dejene Derseh Abateneh, Melkamu Beyene Teferi, Abyot Asres.

**Resources:** Dejene Derseh Abateneh, Melkamu Beyene Teferi.

**Software:** Kindie Mitiku Kebede, Melkamu Beyene Teferi, Abyot Asres.

**Supervision:** Kindie Mitiku Kebede, Dejene Derseh Abateneh, Abyot Asres.

**Validation:** Dejene Derseh Abateneh, Abyot Asres.

**Visualization:** Melkamu Beyene Teferi.

**Writing – original draft:** Kindie Mitiku Kebede, Dejene Derseh Abateneh.

**Writing – review & editing:** Kindie Mitiku Kebede, Dejene Derseh Abateneh, Melkamu Beyene Teferi, Abyot Asres.

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
