## [Editor Report · Decision Letter 0]

22 Jun 2020

PONE-D-20-08526

Chronic Kidney Disease and Associated factors among adult population in Southwest Ethiopia

PLOS ONE

Dear Dr. Abateneh,

Thank you for submitting your manuscript to PLOS ONE. After careful consideration, we feel that it has merit but does not fully meet PLOS ONE’s publication criteria as it currently stands. Therefore, we invite you to submit a revised version of the manuscript that addresses the points raised during the review process.

We recommend to include the potential impact of the kebeles in the results of the survey.  

We requiere to supress these first lines and start the paper directly with the CKD.

Authors are required to re-define what advanced age means or modify the age ranges. > 40 years cannot be considered advanced age

The increased prevalence of CKD in the current study remains unexplained and scarcely discussed. Authors are requiered to recommend the use of CKD-EPI, MDRD or CG to estimate CKD prevalence in order to make recommendations.

The paper is suitable for publication but needs to focus in the novelty and originality of the studied population. We do not identify  any conflicts of interest.

We look forward to receiving your revised manuscript.

Kind regards,

Domingo Gonzalez-Lamuño, Ph.D:, M.D.

Academic Editor

PLOS ONE

Additional Editor Comments:

There are similar papers as Sumaili et al in 2009 about prevalence of Chronic Kidney Disease in the Democratic Republic of Congo. Although there is no significant new clinical data, the study is based in a survey and there are some socio demographic data that are of interest. The survey is based in an administrative Ethiopian distribution of kebeles, making it original. However these points are not discussed along the paper. It would be of interest to compare the urban and rural areas.

Moreover, the introduction is initially focused in the global burden of noncommunicable diseases (NCD) but this not the main point along the paper.

Hypertension and advanced age were significantly associated with the excess risk of CKD, however the advanced age criteria is > 40 years. Authors must re-define what advanced age means or modify the age ranges.

The increased prevalence of CKD in the current study remains unexplained and scarcely discussed. Authors must recommend the use of CKD-EPI, MDRD or CG to estimate CKD prevalence in order to make recommendations.

The paper is suitable for publication but needs to focus in the novelty and originality of the studied population. There are no conflicts of interest.

I consider that after this corrections in the introduction (eliminate the initial lines related to NCD, and start with CKD prevalence and the collected data related with general and CKD populations

Journal Requirements:

2. If you developed a questionnaire as part of this study and it is not under a copyright more restrictive than CC-BY, please include a copy, in both the original language and English, as Supporting Information.
---

## [Author Response · Author response to Decision Letter 0]

27 Jul 2020

Ref No: PONE-D-20-08526

Title: Chronic Kidney Disease and Associated Factors among Adult Population in Southwest Ethiopia

Response to reviewers

1. We recommend to include the potential impact of the kebeles in the results of the survey. 

Response: Dear reviewer, all the kebeles included in this study were urban kebeles and assumed to be homogenous. Therefore, the prevalence of CKD is not expected to vary among the included kebeles.

2. We require to suppress these first lines and start the paper directly with the CKD.

Response: We have deleted the first paragraph and started the paper directly with the CKD (Introduction section, page 2, line 38-44). 

3. Authors are required to re-define what advanced age means or modify the age ranges. > 40 years cannot be considered advanced age

Response: Dear reviewer thank you very much for this comment. We couldn’t find a standard definition for advanced age for sub-Saharan countries. For this reason, we have removed the term “advanced age” and simply we categorized age as ≥40 years old and <40 years old. The age category was decided for comparison (Compare with other studies) (Abstract section, page 2, line 30; Discussion section, page 25, line 316; Conclusions section, page 26, line 337and 341). 

4. The increased prevalence of CKD in the current study remains unexplained and scarcely discussed. Authors are required to recommend the use of CKD-EPI, MDRD or CG to estimate CKD prevalence in order to make recommendations.

Response: Even though the aim of this study is not to recommend the use of CKD-EPI, MDRD or CG, we have incorporated recommendations on the use of CKD-EPI, MDRD, or CG to estimate CKD prevalence in this study area (Abstract section, page 2, line 27-28).

5. The paper is suitable for publication but needs to focus in the novelty and originality of the studied population. 

Response: Dear reviewer, thank you for this comment. At this stage, we can’t change the study population. It is difficult to say that the study population is unique. This study was conducted among the urban population in the Sheka zone. Therefore, the population may share similar characteristics with other urban populations in Ethiopia. However, this study is the first community-based study in the study area and in Ethiopia at large. In fact, this study has certain limitations. We have included those limitations in the manuscript. 

Response to Editor

1. There are similar papers as Sumaili et al in 2009 about prevalence of Chronic Kidney Disease in the Democratic Republic of Congo. Although there is no significant new clinical data, the study is based in a survey and there are some socio-demographic data that are of interest. The survey is based in an administrative Ethiopian distribution of kebeles, making it original. However, these points are not discussed along the paper.

Response: Dear editor, all the kebeles included in this study were urban kebeles and assumed to be homogenous. Therefore, the prevalence of CKD is not expected to vary among the included kebeles.

2. It would be of interest to compare the urban and rural areas.

Response: Dear editor, this study was exclusively studied in urban areas. Therefore, it is difficult to compare urban and rural areas.

3. Moreover, the introduction is initially focused in the global burden of noncommunicable diseases (NCD) but this not the main point along the paper.

Response: We have deleted the first paragraph and started the paper directly with the CKD (page 2, line 38-44).

4. Hypertension and advanced age were significantly associated with the excess risk of CKD, however the advanced age criteria is > 40 years. Authors must re-define what advanced age means or modify the age ranges.

Response: Dear editor, thank you very much for this comment. We couldn’t find a standard definition for advanced age for sub-Saharan countries. For this reason, we removed the term “advanced age” and simply we categorized age as ≥ 40 years old and <40 years old. The age category was decided for comparison (Compare with other studies) (Abstract section, page 2, line 30; Discussion section, page 25, line 316; Conclusions section, page 26, line 337 and 341). 

5. The increased prevalence of CKD in the current study remains unexplained and scarcely discussed. Authors must recommend the use of CKD-EPI, MDRD or CG to estimate CKD prevalence in order to make recommendations.

Response: Even though the aim of this study is not to recommend the use of CKD-EPI, MDRD, or CG, we have incorporated recommendation on use of CKD-EPI, MDRD or CG to estimate CKD prevalence in this study area (page 2, line 27-28)

6. The paper is suitable for publication but needs to focus in the novelty and originality of the studied population. 

Response: Dear editor, thank you for this comment. At this stage, we can’t change the study population. It is difficult to say that the study population is unique. This study was conducted among the urban population in the Sheka zone. Therefore, the population may share similar characteristics with other urban populations in Ethiopia. However, this study is the first community-based study in the study area and in Ethiopia at large. In fact, this study has certain limitations. We have included those limitations in the manuscript. 

7. I consider that after this correction in the introduction (eliminate the initial lines related to NCD, and start with CKD prevalence and the collected data related with general and CKD populations

 Response: Corrected based on your comment. 

8. If you developed a questionnaire as part of this study and it is not under a copyright more restrictive than CC-BY, please include a copy, in both the original language and English, as Supporting Information.

Response: We have uploaded a questionnaire both in English and local language (Amharic) as supporting information.

---

## [Decision Letter · Decision Letter 1]

30 Jun 2021

PONE-D-20-08526R1

Chronic Kidney Disease and Associated Factors among Adult Population in Southwest Ethiopia

PLOS ONE

Dear Dr. Derseh,

Thank you for submitting your manuscript to PLOS ONE. After careful consideration, we feel that it has merit but does not fully meet PLOS ONE’s publication criteria as it currently stands. Therefore, we invite you to submit a revised version of the manuscript that addresses the points raised during the review process.

The reviewers have identified several aspects of your study design that require further clarification. Please ensure that you attend carefully to each of their queries when preparing a revised version of your manuscript.

We look forward to receiving your revised manuscript.

Kind regards,

Jamie Males

Staff Editor

PLOS ONE

Journal Requirements:

Additional Editor Comments (if provided):

Authors respond to all the comments of both reviewers and editor. All significant points are corrected in the final version and the paper is suitable for being puiblished. Although the novelty of the resultas are scarce, the study is well designed and the methodology have some interest for the region.

Reviewers' comments:

Reviewer's Responses to Questions

**Comments to the Author**

1. If the authors have adequately addressed your comments raised in a previous round of review and you feel that this manuscript is now acceptable for publication, you may indicate that here to bypass the “Comments to the Author” section, enter your conflict of interest statement in the “Confidential to Editor” section, and submit your "Accept" recommendation.

Reviewer #1: (No Response)

Reviewer #2: All comments have been addressed

2. Is the manuscript technically sound, and do the data support the conclusions?

Reviewer #1: Partly

Reviewer #2: Yes

3. Has the statistical analysis been performed appropriately and rigorously? 

Reviewer #1: Yes

Reviewer #2: Yes

4. Have the authors made all data underlying the findings in their manuscript fully available?

Reviewer #1: Yes

Reviewer #2: No

5. Is the manuscript presented in an intelligible fashion and written in standard English?

Reviewer #1: Yes

Reviewer #2: Yes

6. Review Comments to the Author

Reviewer #1: 1. The work is original and will contribute to body of knowledge

2. The author should pay attention and correction the minor edits in the manuscript

3. Figures should carry units

4. Presentation of tables should not be merged together like tables 6-8. Each should be presented separately, highlighting what is significant and relevant to the study.

5. Discussions should focus more on the findings based on the objectives of the study

6. The conclusion should be derived from the outcomes or findings of the study and should not be inferred or extrapolated.

Reviewer #2: 1. On page 13, row 229, please clarify the sentence “As the outcome variable is dictums, we applied logistic regression model”.

2. Please indicate in the methods how was central obesity defined. The methods seem to define obesity only and waist circumference, which is not the same as central obesity. The latter is habitually defined as waist to height ratio.

3. There are a couple places in the results where the statement “developed chronic renal failure” is being used. Since this is a cross-sectional study, time-dependent terminology should be avoided. The authors may wish to use something on the line of “had chronic renal failure” instead.

4. The lack of associations with other risk factors than HT might be due to the low prevalence of these other factors, i.e. CVD, diabetes, in the studied population. The authors may wish to comment on this aspect.

7. PLOS authors have the option to publish the peer review history of their article (what does this mean?). If published, this will include your full peer review and any attached files.

Reviewer #1: **Yes: **Timothy Olusegun Olanrewaju

Reviewer #2: No

---

## [Author Response · Author response to Decision Letter 1]

17 Aug 2021

Ref No: PONE-D-20-08526R1

Title: Chronic Kidney Disease and Associated Factors among Adult Population in Southwest Ethiopia

Response to reviewers

Reviewer #1: 

1. The work is original and will contribute to body of knowledge

Response: Dear reviewer, thanks for your remark.

2. The author should pay attention and correction the minor edits in the manuscript

Response: Dear Reviewer, thanks for your comment and we have corrected the minor edits within the manuscript accordingly.

3. Figures should carry units

Response: We have added the units with figures accordingly (Abstract section, page 2, line 26-27, Result section, page 13, line 217, 229 ).

4. Presentation of tables should not be merged together like tables 6-8. Each should be presented separately, highlighting what is significant and relevant to the study.

Response: We have cited the tables separately (Result section, page 14, line 251).

5. Discussions should focus more on the findings based on the objectives of the study

Response: Dear reviewer, we appreciate your comment. In the discussion section, we tried to be focused based on the two specific objectives of the study. i. the prevalence of CKD based on the GFR estimators and ii. associated factors of CKD. The discussion is based on the finding of the two objectives.

6. The conclusion should be derived from the outcomes or findings of the study and should not be inferred or extrapolated.

Response: Dear reviewer, we appreciate your comment and we have corrected the conclusion section mainly on the conclusion on the prevalence of CKD and avoid extrapolation during the conclusion (Conclusions, page 26, line 339).

Reviewer #2: 

1. On page 13, row 229, please clarify the sentence “As the outcome variable is dictums, we applied logistic regression model”.

Response: Thanks for your comment. The sentence “As the outcome variable is dictums, we applied logistic regression model” is to means “As the outcome variable is dichotomous, we applied logistic regression model.” We re-wrote the sentence. It’s to indicate the outcome variable CKD has one of two possible values and fulfills one of the logistic regression assumptions (Result section, page 14, line 232). 

2. Please indicate in the methods how was central obesity defined. The methods seem to define obesity only and waist circumference, which is not the same as central obesity. The latter is habitually defined as waist to height ratio.

Response: Thanks for your comment and keen observation. The waist circumference is used to assess central obesity and it’s indicated in previous studies [1, 2]. The waist circumference in the current study is used to define central obesity with the cut-point for male ≥102 cm and female ≥88cm. We have corrected the statement accordingly in the definition of variables section of the manuscript (Methods section, Variables’ definition, page 6, line 132-133). 

1. Ahmad N, Adam SI, Nawi AM, Hassan MR, Ghazi HF. Abdominal Obesity Indicators: Waist Circumference or Waist-to-hip Ratio in Malaysian Adults Population. Int J Prev Med. 2016 Jun 8;7:82. doi: 10.4103/2008-7802.183654. PMID: 27330688; PMCID: PMC4910307.

2. Lean MEJ, Han TS & Morrison CE (1995): Waist circumference as a measure for indicating the need for weight management. Br. Med. J. 311, 158–161.

 3. There are a couple of places in the results where the statement “developed chronic renal failure” is being used. Since this is a cross-sectional study, time-dependent terminology should be avoided. The authors may wish to use something on the line of “had chronic renal failure” instead.

Response: We appreciate your comment and have used “had chronic renal failure” rather than “developed chronic renal failure” accordingly (Result section, page 12, 13, line 18,222, 225). 

4. The lack of associations with other risk factors than HT might be due to the low prevalence of these other factors, i.e. CVD, diabetes, in the studied population. The authors may wish to comment on this aspect.

Response: Dear Reviewer, we appreciate your comment. We strongly believe that the lack of association between CKD and factors such as Diabetes and CVD in the current study might be the low prevalence or the low number of study participants with diabetes and CVD that result from no association during logistic regression analysis. Chronic diseases such as diabetes and CVD are known risk factors indicated in different studies.

---

## [Decision Letter · Decision Letter 2]

9 Feb 2022

PONE-D-20-08526R2Chronic Kidney Disease and Associated Factors among Adult Population in Southwest EthiopiaPLOS ONE

Dear Dr. Derseh,

Thank you for submitting your manuscript to PLOS ONE. After careful consideration, we feel that it has merit but does not fully meet PLOS ONE’s publication criteria as it currently stands. Therefore, we invite you to submit a revised version of the manuscript that addresses the points raised during the review process.

The article is much improved after addressing the reviewers comments. However, I think that it needs further improvement and incorporation of relevant literature. In this connection, I suggest to cite the following very relevant articles either in Introduction or Discussion section of the manuscript: https://link.springer.com/article/10.1007/s11845-018-1813-2https://bmcpublichealth.biomedcentral.com/articles/10.1186/s12889-019-6796-zhttps://link.springer.com/article/10.1007/s11255-018-1834-9https://link.springer.com/article/10.1186/s40545-019-0169-yhttps://www.hindawi.com/journals/bmri/2016/9710965/

We look forward to receiving your revised manuscript.

Kind regards,

Amjad Khan, Ph.D.

Academic Editor

PLOS ONE

Journal Requirements:

Reviewers' comments:

Reviewer's Responses to Questions

**Comments to the Author**

1. If the authors have adequately addressed your comments raised in a previous round of review and you feel that this manuscript is now acceptable for publication, you may indicate that here to bypass the “Comments to the Author” section, enter your conflict of interest statement in the “Confidential to Editor” section, and submit your "Accept" recommendation.

Reviewer #2: All comments have been addressed

2. Is the manuscript technically sound, and do the data support the conclusions?

Reviewer #2: Yes

3. Has the statistical analysis been performed appropriately and rigorously? 

Reviewer #2: Yes

4. Have the authors made all data underlying the findings in their manuscript fully available?

Reviewer #2: (No Response)

5. Is the manuscript presented in an intelligible fashion and written in standard English?

Reviewer #2: Yes

6. Review Comments to the Author

Reviewer #2: Kindie Mitiku Kebede et al. studied the burden of CKD and associated risk factors among adults in Southwest Ethiopia for 2018. They found a prevalence of CKD that was comparable with other sub-Saharan African countries, with hypertension and age ≥40 years old being significant risk factors for CKD. The authors have responded satisfactorily to this reviewer's concerns. I have no further comments.

7. PLOS authors have the option to publish the peer review history of their article (what does this mean?). If published, this will include your full peer review and any attached files.

Reviewer #2: No

---

## [Author Response · Author response to Decision Letter 2]

14 Feb 2022

RESPONSE TO EDITOR AND REVIEWER COMMENTS

Dear Editor and Reviewers,

The authors sincerely appreciate the time and resources invested in giving such a thorough review. Thank you for the previous comments, suggestions and corrections on the manuscript. We have made efforts to address them and revise the manuscript in line with the comments. The changes have been highlighted in red and tracked in the manuscript. We look forward to a favorable and kind re-consideration.

Editor comment

The article is much improved after addressing the reviewers comments. However, I think that it needs further improvement and incorporation of relevant literature. In this connection, I suggest to cite the following very relevant articles either in Introduction or Discussion section of the manuscript:

https://link.springer.com/article/10.1007/s11845-018-1813-2

https://bmcpublichealth.biomedcentral.com/articles/10.1186/s12889-019-6796-z

https://link.springer.com/article/10.1007/s11255-018-1834-9

https://link.springer.com/article/10.1186/s40545-019-0169-y

https://www.hindawi.com/journals/bmri/2016/9710965/

Response: Dear Editor, thank you for your insights and the suggested articles to be incorporated within the manuscript. The articles are very interesting. We have added a reference that supports a statement in the discussion section (Ref#30, Page 25, Line 29). However, the other articles are studies on hemodialysis patients (focuses on depression among hemodialysis patients, hypertension control among hemodialysis patients, management of patient care in hemodialysis, and on factors affecting to achieve dry weight in post-dialysis) and we couldn’t cite as our study mainly focuses on the prevalence of CKD through a community-based cross-sectional study. But we will use it for other studies in the future.

---

## [Editor Report · Decision Letter 3]

15 Feb 2022

Chronic Kidney Disease and Associated Factors among Adult Population in Southwest Ethiopia

PONE-D-20-08526R3

Dear Dr. Derseh,

We’re pleased to inform you that your manuscript has been judged scientifically suitable for publication and will be formally accepted for publication once it meets all outstanding technical requirements.

Kind regards,

Amjad Khan, Ph.D.

Academic Editor

PLOS ONE

Additional Editor Comments (optional):

Thank you for considering and addressing my comments.
---

## [Editor Report · Acceptance letter]

24 Feb 2022

PONE-D-20-08526R3 

Chronic Kidney Disease and Associated Factors among Adult Population in Southwest Ethiopia 

Dear Dr. Abateneh:

I'm pleased to inform you that your manuscript has been deemed suitable for publication in PLOS ONE. Congratulations! Your manuscript is now with our production department. 

Kind regards, 

on behalf of

Dr. Amjad Khan 

Academic Editor

PLOS ONE